# Composed Image Retrieval with Text Feedback via Multi-grained Uncertainty Regularization

**Yiyang Chen** [1,2*]  **Zhedong Zheng**[3†]  **Wei Ji**[1]  **Leigang Qu**[1]  **Tat-Seng Chua**[1]

[1] Sea-NExT Joint Lab, National University of Singapore   [2] Tsinghua University
[3] Faculty of Science and Technology, and Institute of Collaborative Innovation, University of Macau

## Abstract

We investigate composed image retrieval with text feedback. Users gradually look for the target of interest by moving from coarse to fine-grained feedback. However, existing methods merely focus on the latter, *i.e.*, fine-grained search, by harnessing positive and negative pairs during training. This pair-based paradigm only considers the one-to-one distance between a pair of specific points, which is not aligned with the one-to-many coarse-grained retrieval process and compromises the recall rate. In an attempt to fill this gap, we introduce a unified learning approach to simultaneously modeling the coarse- and fine-grained retrieval by considering the multi-grained uncertainty. The key idea underpinning the proposed method is to integrate fine- and coarse-grained retrieval as matching data points with small and large fluctuations, respectively. Specifically, our method contains two modules: uncertainty modeling and uncertainty regularization. (1) The uncertainty modeling simulates the multi-grained queries by introducing identically distributed fluctuations in the feature space. (2) Based on the uncertainty modeling, we further introduce uncertainty regularization to adapt the matching objective according to the fluctuation range. Compared with existing methods, the proposed strategy explicitly prevents the model from pushing away potential candidates in the early stage, and thus improves the recall rate. On the three public datasets, *i.e.*, FashionIQ, Fashion200k, and Shoes, the proposed method has achieved +4.03%, +3.38%, and +2.40% Recall@50 accuracy over a strong baseline, respectively.

## 1 Introduction

Despite the great success of recent deeply-learned retrieval systems (Radenović et al., 2018; Zheng et al., 2017a; 2020b), obtaining accurate queries remains challenging. Users usually cannot express clearly their intentions at first glance or describe the object of interest with details at the very beginning. Considering such an obstacle, the retrieval system with feedback is preferable since it is similar to the human recognition process, which guides the users to provide more discriminative search conditions. These conditions are used to narrow down the search scope effectively and efficiently, such as *"I want similar shoes but with red color"*. In this work, without loss of generality, we study a single-round real-world scenario, *i.e.*, composed image retrieval with text feedback. This task is also named text-guided image retrieval. Given one query image and one feedback, the retrieval model intends to spot the image, which is similar to the query but with modified attributes according to the text. Such an ideal text-guided image retrieval system can be widely applied to shopping and tourism websites to help users find the target products / attractive destinations without the need to express a clear intention at the start (Liu et al., 2012). Recent developments in this field are mainly attributed to two trends: 1) the rapid development of deeply-learned methods in both computer vision and natural language processing communities, *e.g.*, the availability of pre-trained large-scale cross-modality models, such as CLIP (Radford et al., 2021); and 2) the fine-grained metric learning in biometric recognition, such as triplet loss with hard mining (Hermans et al., 2017; Oh Song et al., 2016), contrastive loss (Zheng et al., 2017b), and infoNCE loss (Oord et al., 2018; Han et al., 2022), which mines one-to-one relations among pair- or triplet-wise inputs.

---

*Work done during an internship at NUS. † Correspondence to zhedongzheng@um.edu.mo.

However, one inherent problem remains: how to model the coarse-grained retrieval? The fine-grained metric learning in biometrics *de facto* is designed for strict one-to-one fine-grained matching, which is not well aligned with the multi-grained image retrieval with text feedback. As shown in Figure 1a, we notice that there exists multiple true matchings or similar candidates. If we still apply pair-wise metric learning, it will push away these true positives, compromising the training on coarse-grained annotations.

As an attempt to loosen this restriction, we introduce a matching scheme to model an uncertainty range, which is inspired by the human retrieval process from coarse-grained range to fine-grained range. As shown in Figure 1b, we leverage uncertain fluctuation to build the multi-grained area in the representation space. For the detailed query, we still apply one-to-one matching as we do in biometric recognition. On the other hand, more common cases are one-to-many matching. We conduct matching between one query point and a point with an uncertain range. The uncertain range is a feature space, including multiple potential candidates due to the imprecise query images or the ambiguous textual description. Jointly considering the two kinds of query, we further introduce a unified uncertainty-based approach for both one-to-one matching and one-to-many matching, in the multi-grained image retrieval with text feedback. In particular, the unified uncertainty-based approach consists of two modules, *i.e.*, uncertainty modeling and uncertainty regularization. The uncertainty modeling simulates the real uncertain range within a valid range. The range is estimated based on the feature distribution within the mini-batch. Following the uncertainty modeling, the model learns from these noisy data with different weights. The weights are adaptively changed according to the fluctuation range as well. In general, we will not punish the model, if the query is ambiguous. In this way, we formulate the fine-grained matching with the coarse-grained matching in one unified optimization objective during training. Different from only applying one-to-one matching, the uncertainty regularization prevents the model from pushing away potential true positives, thus improving the recall rate. Our contributions are as follows.

- We pinpoint a training/test misalignment in real-world image retrieval with text feedback, specifically between fine-grained metric learning and the practical need for coarse-grained inference. Traditional metric learning mostly focuses on one-to-one alignment, adversely impacting one-to-many coarse-grained learning. This problem identification underscores the gap we address in our approach.
- We introduce a new unified method to learn both fine- and coarse-grained matching during training. In particular, we leverage the uncertainty-based matching, which consists of uncertainty modeling and uncertainty regularization.
- Albeit simple, the proposed method has achieved competitive recall rates, *i.e.*, 61.39%, 79.84% and 70.2% Recall@50 accuracy on three large-scale datasets, *i.e.*, FashionIQ, Fashion200k, and Shoes, respectively. Since our method is orthological to existing methods, it can be combined with existing works to improve performance further.

## 2 RELATED WORK

**Composed Image Retrieval with Text Feedback.** Traditional image retrieval systems utilize one relevant image as a query (Philbin et al., 2007; Zheng et al., 2020a). It is usually challenging to acquire such an accurate query image in advance. The multimodal query involves various input queries of different modalities, such as image and text, which eases the difficulty in crafting a query and provides more diverse details. In this work, we focus on image retrieval with text feedback, also called text-guided image retrieval. Specifically, the input query contains a reference image and textual feedback describing the modifications between the reference image and the target image. The critical challenge is how to compose the two modality inputs properly (He et al., 2022; Yang et al., 2016; Saito et al., 2023; Shin et al., 2021), and align visual and language space (Radford et al., 2021; Norelli et al., 2022; Saito et al., 2023). The existing methods (Lee et al., 2021; Baldrati et al., 2022) usually extract the textual and visual features of the query separately through the text encoder and image encoder. These two types of features are composited as the final query embeddings to match the visual features of the target image. Generally, there are two families of works on image retrieval with text feedback based on whether using the pre-trained model. The first line of work mainly studies how to properly combine the features of the two modalities (Qu et al., 2021; Han et al., 2022). Content-Style Modulation (CosMo) (Lee et al., 2021) proposes an image-based compositor containing two independent modulators. Given the visual feature of the reference image,

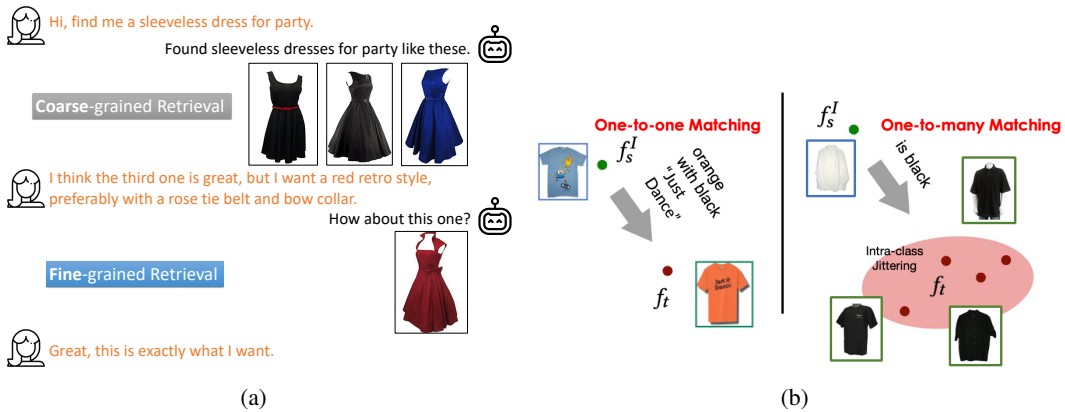

(a)                                                                (b)

Figure 1: (a) The typical retrieval process contains two steps, *i.e.*, the coarse-grained retrieval and fine-grained retrieval. The coarse-grained retrieval harnesses the brief descriptions or imprecise query images, while the fine-grained retrieval requires more details for one-to-one mapping. The existing approaches usually focus on optimizing the strict pair-wise distance during training, which is different from the one-to-many coarse-grained test setting. Overwhelming one-to-one metric learning compromises the model to recall potential candidates. (b) Our intuition. We notice that there exist two typical matching types for the fine- and coarse-grained retrieval. Here we show the difference between one-to-one matching (left) and one-to-many matching (right).

the content modulator first performs local updates on the feature map according to text features. The style modulator then recovers the visual feature distribution with the original feature mean and std for matching. Furthermore, CLVC-Net (Wen et al., 2021) introduces two fine-grained compositors: a local-wise image-based compositor and a global-wise text-based compositor. Following the spirit of mutual learning (Zhang et al., 2018), two compositors are learned from each other considering the prediction consistency. With the rapid development of the big model, another line of work is to leverage the model pre-trained on large-scale cross-modality datasets, and follow the pretrain-finetune paradigm. For instance, CLIP4Cir (Baldrati et al., 2022) applies CLIP (Radford et al., 2021) as the initial network to integrate text and image features, and adopts a two-stage training strategy to ease the optimization. Similar to CLIP, CLIP4Cir fine-tunes the CLIP text encoder and CLIP visual encoder for feature matching in the first stage. In the second stage, the two encoders are fixed and a non-linear compositor is added to fuse the text and visual feature in an end-to-end manner. Taking one step further, Zhao et al. (2022) introduce extra large-scale datasets, *e.g.*, FACAD (Yang et al., 2020), FashionGen (Rostamzadeh et al., 2018), for pretraining. Different from these existing works, in this work, we do not focus on the network structure or pretraining weights. Instead, we take a closer look at the multi-grained matching, especially the coarse-grained one-to-many matching during training. We explicitly introduce the uncertainty range to simulate the intra-class jittering.

**Uncertainty Learning.** With the rapid development of data-driven methods, the demands on the model reliability rise. For instance, one challenging problem still remains how to measure the "confidence" of a prediction. Therefore, some researchers resort to uncertainty. Kendall & Gal (2017) divide the uncertainty into two major categories, *i.e.*, epistemic uncertainty and aleatoric uncertainty. The former epistemic uncertainty denotes model uncertainty that the models, even trained on the same observation (dataset), learn different weights. The typical work of this direction is Bayesian networks (Jordan et al., 2007; Gal & Ghahramani, 2016), which does not learn any specific weights but the distribution of weights. In a similar spirit, Monte Carlo Dropout (Gal & Ghahramani, 2016) is proposed to simulate the Bayesian networks during inference, randomly dropping the network weights. Another family of works deals with the inherent data uncertainty, usually caused by device deviations or ambiguous annotations. This line of uncertainty approaches has been developed in many fields, including image retrieval (Warburg et al., 2021), image classification (Postels et al., 2021), image segmentation (Zheng & Yang, 2021), 3D reconstruction (Truong et al., 2021), and person re-id (Yu et al., 2019; Zhang et al., 2022). In terms of representation learning, (Chun et al., 2021; Pishdad et al., 2022) directly regress the mean and variance of the input data and use the probability distribution similarity instead of the cosine similarity. Similar to (Chun et al., 2021; Pishdad et al., 2022), Oh et al. (2018) use Monte Carlo sampling to obtain the averaged distribution similarity. Besides, Warburg et al. (2021) directly consider the loss variance instead of the feature

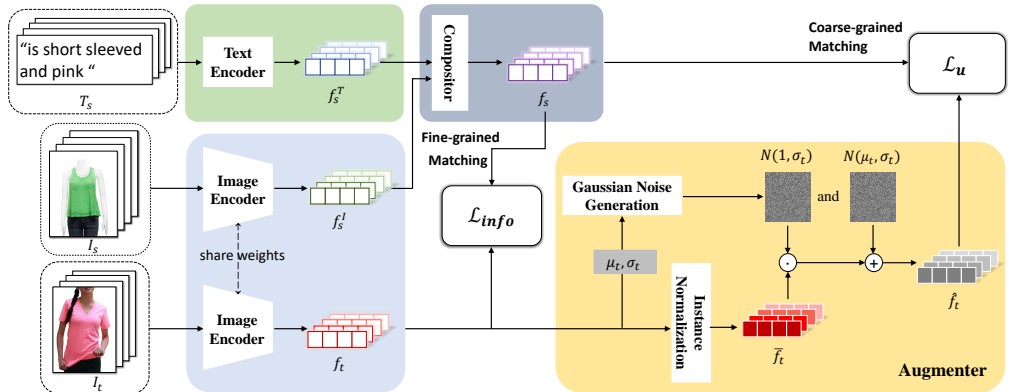

Figure 2: The overview of our network. Given the source image $I_s$ and the text $T_s$ for modification, we obtain the composed features $f_s$ by combining $f_s^T$ and $f_s^I$ via compositor. The compositor contains a content module and a style module. Meanwhile, we extract the visual features $f_t$ of the target image $I_t$ via the same image encoder as the source image. Our main contributions are the uncertainty modeling via augmenter, and the uncertainty regularization for coarse matching. (1) The proposed augmenter applies feature-level noise to $f_t$, yielding $\hat{f}_t$ with identical Gaussian Noise $N(1, \sigma_t)$ and $N(\mu_t, \sigma_t)$, respectively. Albeit simple, it is worth noting that the augmented feature $\hat{f}_t$ simulates the intra-class jittering of the target image, following the original feature distribution. (2) The commonly used InfoNCE loss focuses on the fine-grained one-to-one mapping between the original target feature $f_t$ and the composited feature $f_s$. Different from InfoNCE loss, the proposed method harnesses the augmented feature $\hat{f}_t$ and $f_s$ to simulate the one-to-many mapping, considering different fluctuations during training. Our model applies both the fine-grained matching and the proposed coarse-grained uncertainty regularization, facilitating the model training.

variance. In particular, they re-formulate the original triplet loss to involve the consideration of loss variance. Another line of works (Chang et al., 2020; Dou et al., 2022) directly adds noise to features to simulate the uncertainty. The variance is also from the model prediction like (Chun et al., 2021; Pishdad et al., 2022) and a dynamic uncertainty-based loss is deployed as (Kendall & Gal, 2017; Zheng & Yang, 2021). Similarly, in this work, we also focus on enabling the model learning from multi-grained annotations, which can be viewed as an annotation deviation. Differently, there are two primary differences from existing uncertainty-based works: (1) We explicitly introduce Gaussian Noise to simulate the data uncertainty in terms of the coarse-grained retrieval. For instance, we simulate the "weak" positive pairs by adding intra-class jittering. (2) We explicitly involve the noisy grade in optimization objectives, which unifies the coarse- and fine-grained retrieval. If we add zero noise, we still apply the strict one-to-one metric learning objective. If we introduce more noise, we give the network less punishments. It is worth noting that unifying the fine- and coarse-grained retrieval explicitly preserve the original representation learning, while preventing over-fitting to ambiguous coarse-grained samples in the dataset.

## 3 METHOD

### 3.1 PROBLEM DEFINITION

We show the brief pipeline in Figure 2. **In this paper, we do not pursue a sophisticated network structure but a new learning strategy. Therefore, we mainly follow existing works (Lee et al., 2021; Chen et al., 2020) to build the network for a fair comparison, and more structure details are provided in Implementation Details.** Given a triplet input, *i.e.*, one source image input $I_s$, one text sentence $T_s$ and the target image $I_t$, the model extracts the visual feature $f_s^I$ of $I_s$, the target feature $f_t$ of $I_t$ and the textual feature $f_s^T$ of $T_s$. Then we leverage a simple compositor to combine the source visual feature $f_s^I$ and the source text feature $f_s^T$ as the composed feature $f_s$. We intend to increase the cosine similarity between $f_s$ and $f_t$ in the representation space. During test time, we extract $f_s$ as the query feature to find the most similar $f_t$ in the candidate pool.

We mainly consider the uncertainty in the triplet annotations. As shown in Figure 1a, the dataset usually contains multi-grained matching triplets, considering descriptions and input images. If the

description is ambiguous or the source image is relatively imprecise, the system should return multiple candidate images. In the training phase, if we still apply strong one-to-one matching supervision on such triplet, it will significantly compromise the model to recall potential true matches. Therefore, in this work, we mainly focus on the uncertainty in the triplet matching. If not specified, we deploy uncertainty to denote matching uncertainty.

## 3.2 UNCERTAINTY MODELING

We introduce a noise augmenter to simulate the intra-class jittering. As shown in Figure 1b, instead of strict one-to-one matching, we impel the model to focus on one-to-many matching. Therefore, we need to generate the jittering via augmenter. The augmenter directly works on the final representation space. In particular, the augmenter adds Gaussian Noise of the original feature distribution to the target features $f_t$. The mean $\mu_t$ and standard deviation $\sigma_t$ of the Gaussian Noise are calculated from the original feature $f_t$. We then apply the generated Gaussian noise to the whitened feature. Therefore, the final jittered feature $\hat{f}_t$ can be formulated as follows:

$$\hat{f}_t = \alpha \cdot \bar{f}_t + \beta \tag{1}$$

where $\alpha$ and $\beta$ are the noisy vectors with the same shape as the input target feature, $\alpha \sim N(1, \sigma_t), \beta \sim N(\mu, \sigma_t)$, and $\bar{f}_t$ is whitened feature $\bar{f}_t = \frac{f_t - \mu_t}{\sigma_t}$. We apply the element-wise multiplication to re-scale the input feature, so the Gaussian noise mean of $\alpha$ is set as 1, which is different from $\beta$. It is worth noting that, in this way, we make the feature fluctuate in a limited degree, which is close to the original distribution.

## 3.3 UNCERTAINTY REGULARIZATION

The existing methods usually adopt InfoNCE loss (Goldberger et al., 2004; Movshovitz-Attias et al., 2017; Snell et al., 2017; Gidaris & Komodakis, 2018; Vo et al., 2019a; Lee et al., 2021) which can be viewed as a kind of batch-wise classification loss. It can be simply formulated as:

$$\mathcal{L}_{\text{info}}(f_s, \ f_t) = \frac{1}{B} \sum_{i=1}^{B} -\log \frac{\exp\left(\kappa\left(f_s^i, f_t^i\right)\right)}{\sum_{j=1}^{B} \exp\left(\kappa\left(f_s^i, f_t^j\right)\right)}. \tag{2}$$

Given the composed feature $f_s$ and the target feature $f_t$ of a mini-batch with $B$ samples, InfoNCE loss maximizes the self-similarity $\kappa\left(f_s^i, f_t^i\right)$ while minimizing the similarity with other samples $\kappa\left(f_s^i, f_t^j\right)$ $(i \neq j)$ in the batch. We adopt the cosine similarity as $\kappa$, which is $\kappa(f^i, f^j) = \frac{f^i \cdot f^j}{|f^i||f^j|}$.

We note that InfoNCE loss merely focuses on the one-to-one fine-grained matching. In this work, we intend to unify the fine- and coarse-grained matching. Inspired by the aleatoric uncertainty, we propose the uncertainty regularization. The basic InfoNCE loss is a special case of our loss. Given two types of features $\hat{f}_t$ and $f_s$, our uncertainty regularization can be defined as follow:

$$\mathcal{L}_{\text{u}}\left(f_s, \ \hat{f}_t, \sigma\right) = \frac{\mathcal{L}_{\text{info}}\left(f_s, \ \hat{f}_t\right)}{2\sigma^2} + \frac{1}{2}\log \sigma^2. \tag{3}$$

If the $\sigma$ is a constant, our $\mathcal{L}_{\text{u}}$ regresses to a weighted $\mathcal{L}_{\text{info}}$. The main difference from the InfoNCE loss is that we adaptively tune the optimization objective according to the jittering level in $\hat{f}_t$. If the jittering fluctuation is large (*i.e.*, a large $\sigma$), the weight of first-term InfoNCE loss decreases. In contrast, if the feature has limited changes, the regularization is close to the original InfoNCE loss.

To optimize the multi-grained retrieval performance, we adopt a combination of the fine-grained loss $\mathcal{L}_{\text{inf}}$ and the proposed uncertainty regularization $\mathcal{L}_{\text{u}}$. Therefore, the total loss is as follows:

$$\mathcal{L}_{\text{total}} = \gamma \mathcal{L}_{\text{u}}\left(f_s, \ \hat{f}_t, \sigma_t\right) + (1 - \gamma)\mathcal{L}_{\text{info}}(f_s, \ f_t), \tag{4}$$

where $\gamma$ is a dynamic weight hyperparameter to balance the ratio of the fine- and coarse-grained retrieval. $\sigma_t$ is the standard deviation of $f_t$. **If we ignore the constant term, which does not generate backward gradients,** we could rewrite this loss in a unified manner as:

$$\mathcal{L}_{\text{total}} = \gamma \mathcal{L}_{\text{u}}\left(f_s, \ \hat{f}_t, \sigma_t\right) + (1 - \gamma)\mathcal{L}_{\text{u}}\left(f_s, \ f_t, \frac{1}{\sqrt{2}}\right). \tag{5}$$

During training, we gradually decrease the coarse-grained learning, and increase the fine-grained learning by annealing $\gamma = \exp(-\gamma_0 \cdot \frac{current\_epoch}{total\_epoch})$, where $\gamma_0$ is the initial weight. By setting the

exponential function, we ensure $\gamma \in [0, 1]$. If $\gamma = 0$, we only consider the fine-grained retrieval as existing works (Lee et al., 2021). If $\gamma = 1$, we only consider the coarse-grained one-to-many matching between fluctuated features.

**Discussion. 1). What is the advantage of uncertainty regularization in the feature learning?** The uncertainty regularization is to simulate the one-to-many matching case, which leaves space for multiple ground-truth candidates. It successfully invades the model to over-fitting the strict one-to-one matching as in biometric recognition. As shown in the experiment, the proposed method significantly surpasses other competitive methods in terms of the recall rate, especially Recall@50. **2). How about using uncertainty regularization alone? Why adopt the dynamic weight?** Only coarse matching, which is easy to converge, leads the model to miss the challenging fine-grained matching, even if the description is relatively detailed. Therefore, when the model converges with the "easy" coarse-grained matching during training, we encourage the model to focus back on the fine-grained matching again. The ablation study on $\gamma$ in Section 4 also verifies this point. **3). Extensibility to the Ensembled Dataset.** For instance, in the real-world scenario, one common challenge is how to learn from the ensembled dataset. The ensembled dataset may contain fine-grained text descriptions as well as coarse-grained attribute annotations, like key words. For such a case, the proposed method could, in nature, facilitate the model to learn from multi-grained data. The experiment on the three subsets of FashionIQ verifies the scalability of the proposed method.

## 4 EXPERIMENT

**Implementation Details.** We employ the pre-trained models as our backbone: ResNet-50 (He et al., 2015) on ImageNet as the image encoder and RoBERTa (Liu et al., 2019) as the text encoder. In particular, we adopt the same compositor structure in the existing work CosMo for a fair comparison, which contains two modules: the Content Modulator (CM) and the Style Modulator (SM). The CM uses a Disentangled Multi-modal Non-local block (DMNL) to perform local updates to the reference image feature, while the SM employs a multimodal attention block to capture the style information conveyed by the modifier text. The outputs of the CM and SM are then fused using a simple concatenation operation, and the resulting feature is fed into a fully connected layer to produce the final image-text representation. SGD optimizer (Robbins & Monro, 1951) is deployed with a mini-batch of 32 for 50 training epochs and the base learning rate is $2 \times 10^{-2}$ following Lee et al. (2021). We apply the one-step learning rate scheduler to decay the learning rate by 10 at the 45th epoch. We empirically set $w_1 = 1, w_2 = 1$, which control the scale of augmenter generates Gaussian noise, and the initial balance weight $\gamma_0 = 1$. During inference, we extract the composed feature $f_s$ to calculate the cosine similarity with the feature of gallery images. The final ranking list is generated according to the feature similarity. **Reproducibility.** The code is based on Pytorch (Paszke et al., 2019), and annonymous code is at [1]. We will make our code open-source for reproducing all results.

**Datasets.** Without loss of generability, we verify the effectiveness of the proposed method on the fashion datasets, which collect the feedback from customers easily, including FashionIQ (Wu et al., 2021), Fashion200k (Han et al., 2017) and Shoes (Guo et al., 2018). Each image in these fashion datasets is tagged with descriptive texts as product description, such as "similar style t-shirt but white logo print". **FashionIQ.** We follow the training and test split of existing works (Chen et al., 2020; Lee et al., 2021). Due to privacy changes and deletions, some links are out-of-the-date. We try our best to make up for the missing data by requesting other authors. As a result, we download 75,384 images and use 46,609 images in the original protocol for training. **Shoes.** Shoes (Guo et al., 2018) crawls 10,751 pairs of shoe images with relative expressions that describe fine-grained visual differences. We use 10,000 samples for training and 4,658 samples for evaluation. **Fashion200k.** Fashion200k (Han et al., 2017) has five subsets: *dresses*, *jackets*, *pants*, *skirts*, and *tops*. We deploy 172,000 images for training on all subsets and 33,480 test queries for evaluation. **Evaluation metric.** Following existing works, we report the average Recall@1, Recall@10, and Recall@50 of all queries.

**Comparison with Competitive Methods.** We show the recall rate on FashionIQ in Table 1, including the three subsets and the average score. We could observe four points: (1) The method with uncertainty modeling has largely improved the baseline in both Recall@10 and Recall@50 accuracy, verifying the motivation of the proposed component on recalling more potential candidates.

---

[1] https://github.com/Monoxide-Chen/uncertainty_retrieval

Table 1: Results on FashionIQ. The best performance is in **bold**. Here we show the recall rate R@K, which denotes Recall@K. Average denotes the mean of R@K on all subsets. It is worth noting that our method with one ResNet-50 is competitive with CLIP4Cir (Baldrati et al., 2022) of 4×ResNet-50 in R@50. We train models with different initializations as model ensembles. * Note that FashionViL introduces extra datasets for training.

| Method | Visual Backbone | Dress | | Shirt | | Toptee | | Average | |
|---|---|---|---|---|---|---|---|---|---|
| | | R@10 | R@50 | R@10 | R@50 | R@10 | R@50 | R@10 | R@50 |
| MRN (Kim et al., 2016) | ResNet-152 | 12.32 | 32.18 | 15.88 | 34.33 | 18.11 | 36.33 | 15.44 | 34.28 |
| FiLM (Perez et al., 2018) | ResNet-50 | 14.23 | 33.34 | 15.04 | 34.09 | 17.30 | 37.68 | 15.52 | 35.04 |
| TIRG (Vo et al., 2019b) | ResNet-17 | 14.87 | 34.66 | 18.26 | 37.89 | 19.08 | 39.62 | 17.40 | 37.39 |
| Pic2Word (Saito et al., 2023) | ViT-L/14 | 20.00 | 40.20 | 26.20 | 43.60 | 27.90 | 47.40 | 24.70 | 43.70 |
| VAL (Chen et al., 2020) | ResNet-50 | 21.12 | 42.19 | 21.03 | 43.44 | 25.64 | 49.49 | 22.60 | 45.04 |
| ARTEMIS (Delmas et al., 2022) | ResNet-50 | 27.16 | 52.40 | 21.78 | 54.83 | 29.20 | 43.64 | 26.05 | 50.29 |
| CoSMo (Lee et al., 2021) | ResNet-50 | 25.64 | 50.30 | 24.90 | 49.18 | 29.21 | 57.46 | 26.58 | 52.31 |
| DCNet (Kim et al., 2021) | ResNet-50 | 28.95 | 56.07 | 23.95 | 47.30 | 30.44 | 58.29 | 27.78 | 53.89 |
| FashionViL (Han et al., 2022) | ResNet-50 | 28.46 | 54.24 | 22.33 | 46.07 | 29.02 | 57.93 | 26.60 | 52.74 |
| FashionViL* (Han et al., 2022) | ResNet-50 | 33.47 | 59.94 | 25.17 | 50.39 | 34.98 | 60.79 | 31.21 | 57.04 |
| Baseline | ResNet-50 | 24.80 | 52.35 | 27.70 | 55.71 | 33.40 | 63.64 | 28.63 | 57.23 |
| Ours | ResNet-50 | **30.60** | **57.46** | **31.54** | **58.29** | **37.37** | **68.41** | **33.17** | **61.39** |
| CLVC-Net (Wen et al., 2021) | ResNet-50×2 | 29.85 | 56.47 | 28.75 | 54.76 | 33.50 | 64.00 | 30.70 | 58.41 |
| Ours | ResNet-50×2 | **31.25** | **58.35** | **31.69** | **60.65** | **39.82** | **71.07** | **34.25** | **63.36** |
| CLIP4Cir (Baldrati et al., 2022) | ResNet-50×4 | 31.63 | 56.67 | **36.36** | 58.00 | 38.19 | 62.42 | 35.39 | 59.03 |
| Ours | ResNet-50×4 | **32.61** | **61.34** | 33.23 | **62.55** | **41.40** | **72.51** | **35.75** | **65.47** |

Table 2: Results on Shoes and Fashion200k. We mainly compare ours with single model-based methods (the best performance in red, the second-best in orange). Similar to the phenomenon on FashionIQ, we could observe that the proposed method improves the recall rate over baseline, which is aligned with our intuition on multi-grained matching.

| Method | Shoes | | | Fashion200k | | |
|---|---|---|---|---|---|---|
| | R@1 | R@10 | R@50 | R@1 | R@10 | R@50 |
| MRN(Kim et al., 2016) | 11.74 | 41.70 | 67.01 | 13.4 | 40.0 | 61.9 |
| FiLM(Perez et al., 2018) | 10.19 | 38.89 | 68.30 | 12.9 | 39.5 | 61.9 |
| TIRG(Vo et al., 2019b) | 12.6 | 45.45 | 69.39 | 14.1 | 42.5 | 63.8 |
| VAL(Chen et al., 2020) | 16.49 | 49.12 | 73.53 | 21.2 | 49.0 | 68.8 |
| CoSMo(Lee et al., 2021) | 16.72 | 48.36 | 75.64 | 23.3 | 50.4 | 69.3 |
| DCNet(Kim et al., 2021) | - | 53.82 | 79.33 | - | 46.9 | 67.6 |
| ARTEMIS(Delmas et al., 2022) | 18.72 | 53.11 | 79.31 | 21.5 | 51.1 | 70.5 |
| Baseline | 15.26 | 49.48 | 76.46 | 19.5 | 46.7 | 67.8 |
| Ours | 18.41 | 53.63 | 79.84 | 21.8 | 52.1 | 70.2 |

Especially for Recall@50, the proposed method improves the accuracy from 57.23% to 61.39%. (2) Comparing with the same visual backbone, *i.e.*, one ResNet-50, the proposed method has arrived at a competitive recall rate in terms of both subsets and averaged score. (3) The proposed method with one ResNet-50 is competitive with the ensembled methods, such as CLIP4Cir (Baldrati et al., 2022) with 4× ResNet-50. In particular, ours with one ResNet-50 has arrived at 61.39% Recall@50, surpassing CLIP4Cir (59.03% Recall@50). (4) We train our model ensemble with different initialization, and simply concatenate features, which also surpasses both CLVC-Net (Wen et al., 2021) and CLIP4Cir (Baldrati et al., 2022). We observe similar performance improvement on Shoes and Fashion200k in Table 2. (1) The proposed method surpasses the baseline, yielding 18.41% Recall@1, 53.63% Recall@10, and 79.84% Recall@50 on Shoes, and 21.8% Recall@1, 52.1% Recall@10, and 70.2% Recall@50 on Fashion200k. Especially, in terms of Recall@50 accuracy, the model with uncertainty regularization has obtained +3.38% and +2.40% accuracy increase on Shoes and Fashion200k, respectively. (2) Based on one ResNet-50 backbone, the proposed method is competitive with ARTEMIS (Delmas et al., 2022) in both Shoes and Fashion200k, but ours are more efficient, considering that ARTEMIS needs to calculate the similarity score for every input triplets. (3) Our method also achieves better Recall@50 79.84% than CLVC-Net (Wen et al., 2021) 79.47% with 2× ResNet-50 backbone, and arrives at competitive Recall@10 accuracy on Fashion200k.

**Can Dropout replace the uncertainty regularization?** No. Similar to our method, the dropout function also explicitly introduces the noise during training. There are also two primary differences: (1) Our feature fluctuation is generated according to the original feature distribution, instead of a fixed drop rate in dropout. (2) The proposed uncertainty regularization adaptively punishes the network according to the noise grade. To verify this point, we compare the results between ours and the dropout regularization. In particular, we add a dropout layer after the fully connected layer of the text encoder, which is before the compositor. For a fair comparison, we deploy the baseline ($\mathcal{L}_{\text{info}}$) without modeling uncertainty for evaluation and show the results of different dropout rates in Table 3a. We observe that the dropout does not facilitate the model recalling more true candidates. No matter whether the drop rate is set as 0.2 or 0.5, the performance is close to the baseline without the dropout layer. In contrast, only using the uncertainty regularization ($\mathcal{L}_{\text{u}}$) improves Recall@50 from

Table 3: Ablation studies on the Shoes dataset. The average is (R@10+R@50)/2. (a) Impact of dropout. We compare the impact of the dropout rate and the proposed uncertainty regularization $\mathcal{L}_u$. We could observe that dropout has limited impacts on the final retrieval performance. In contrast, the proposed $\mathcal{L}_u$ shows significant recall accuracy boost. (b) Parameter sensitivity of $w_1, w_2$. The left side of the table shows the R@Ks of different values of $w_1$ when we fix $w_2 = 1$. Meanwhile, the right side of the table shows the R@Ks of different values of $w_2$ when we fix $w_1 = 1$. $w_1 = w_2 = 1$ has the best performance and the same results on two sides. (c) The influence of the initial balance weight $\gamma_0$ on the model. (d) Ablation Study on static $\gamma$ as constant.

(a)

| Uncertainty | Drop Rate | R@10 | R@50 | Average |
|---|---|---|---|---|
| Baseline ($\mathcal{L}_{info}$) | - | 49.48 | 76.46 | 62.97 |
| $\mathcal{L}_{info}$ + Dropout | 0.2 | 49.74 | 76.37 | 63.06 |
| $\mathcal{L}_{info}$ + Dropout | 0.5 | 49.00 | 75.83 | 62.42 |
| Dou et al. (2022) | - | 50.14 | 77.89 | 64.01 |
| Augment Source Feature $f_s^I$ | - | 52.20 | 77.75 | 64.98 |
| Only $\mathcal{L}_u$ | - | 50.83 | 77.41 | 64.12 |
| Ours ($\mathcal{L}_{info} + \mathcal{L}_u$) | - | **53.63** | **79.84** | **66.74** |

(c)

| $\gamma_0$ | R@10 | R@50 | Average |
|---|---|---|---|
| 0.1 | 50.80 | 78.64 | 64.72 |
| 0.5 | 51.20 | 79.12 | 65.76 |
| 1 | **53.63** | **79.84** | **66.74** |
| 2 | 49.66 | 77.63 | 63.65 |
| 3 | 50.83 | 77.55 | 64.19 |
| 5 | 50.14 | 76.32 | 63.23 |
| 10 | 49.91 | 76.43 | 63.17 |
| $+\infty$ (baseline) | 49.48 | 76.46 | 62.97 |

(b)

| Scale | $w_2 = 1, w_1$ | | | $w_1 = 1, w_2$ | | |
|---|---|---|---|---|---|---|
| | R@10 | R@50 | Average | R@10 | R@50 | Average |
| 0.1 | 48.42 | 75.57 | 62.00 | 51.03 | 79.10 | 65.07 |
| 0.2 | 51.35 | 76.35 | 63.85 | 50.49 | 79.15 | 64.82 |
| 0.5 | 51.03 | 78.84 | 64.94 | 50.69 | 77.75 | 64.22 |
| 0.7 | 51.95 | 78.95 | 65.45 | 50.43 | 78.38 | 64.41 |
| 1 | **53.63** | **79.84** | **66.74** | **53.63** | **79.84** | **66.74** |
| 2 | 48.71 | 76.80 | 62.76 | 49.83 | 78.41 | 64.12 |
| 5 | 51.29 | 78.06 | 64.68 | 48.20 | 76.83 | 62.52 |
| 7 | 48.91 | 76.23 | 62.57 | 48.60 | 77.18 | 62.89 |
| 10 | 48.71 | 76.80 | 62.76 | 47.02 | 74.71 | 60.87 |

(d)

| $\gamma$ | R@10 | R@50 | Average |
|---|---|---|---|
| 0.2 | 30.90 | 63.12 | 47.01 |
| 0.5 | 41.87 | 73.11 | 57.49 |
| 0.8 | 41.87 | 72.39 | 57.13 |

76.46% to 77.41%. Combining with the fine-grained matching baseline, the uncertainty regularization even improves Recall@10 by +4.15% accuracy. Besides, we re-implement (Dou et al., 2022) by adding two extra branches to regress the feature mean and variance for loss calculation. It has achieved 50.14% Recall@10 and 77.89 % Recall@50, which is inferior to our method.

**Impact of the noise fluctuation.** We study the impact of the noise fluctuation in the uncertainty modeling. In particular, we change the noise scale in Eq. 1 by adjusting the scale of $\alpha$ and $\beta$. The modified formulation is as follows:

$$\hat{f}_t = \alpha' \bar{f}_t + \beta' \tag{6}$$

where $\alpha' \in N(1, w_1\sigma)$ and $\beta' \in N(\mu, w_2\sigma)$. $w_1, w_2$ are the scalars to control the noise scale. We show the impact of different noise scales in Table 3b. We fix one of the $w$ to 1 and change the another parameter. We could observe two points: (1) As we expected, the identical noise setting $(w_1 = 1, w_2 = 1)$ achieves the best performance. This is because such noise is close to the original feature distribution and simulates the fluctuation in the training set. (2) The experiment results also show that our uncertainty regularization can tolerate large amplitude noise changes. Even if the training data contains lots of noise, the network is still robust and achieves reasonable performance. It is attributed to uncertainty regularization that punishes the network according to the noise grade.

**Parameter sensitivity of the balance weight $\gamma$.** As shown in Eq. 4, $\gamma$ is a dynamic weight to help balance the fine- and coarse-grained retrieval. During training, we encourage the model to gradually pay more attention to the fine-grained retrieval. According to $\gamma = \exp(-\gamma_0 \cdot \frac{current\_epoch}{total\_epoch})$, we set different initial values of $\gamma_0$ to change the balance of the two loss functions. We evaluate the model on the Shoes dataset and report results in Table 3c. If $\gamma_0$ is close to 0, the model mostly learns the uncertainty loss on the coarse-grained retrieval, and thus recall rate is still high. In contrast, if $\gamma_0$ is close to $+\infty$, the model only focuses on the fine-grained learning, and thus the model converges to the baseline. Therefore, when deploying the model to the unseen environment, $\gamma_0 = 1$ can be a good initial setting. Besides, results with fixed $\gamma$ are shown in Table 3d. The constant uncertainty loss drives the model to focus on coarse-grained matching, resulting in low recall rates as well.

**Qualitative visualization.** We show the top-5 retrieval results on FashionIQ, Fashion200k, and Shoes in Figure 3. (1) Compared with the baseline, our model captures more fine-grained keywords, like "shiner". (2) The proposed method also recalls more candidate images with a consistent style. It also reflects that the proposed method is robust and provides a better user experience, since most websites display not only top-1 but also top-5 products.

**Training Convergence.** As shown in Table 4a, the baseline model (blue line) is prone to overfit all labels, including the coarse-grained triplets. Therefore, the training loss converges to zero quickly.

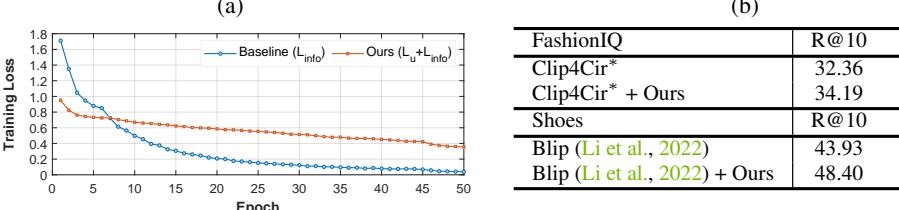

Figure 3: Qualitative image retrieval result on FashionIQ, Fashion200k and Shoes. We mainly compare the top-5 ranking list of the proposed method with the baseline. (Please zoom in.)

Table 4: (a) The training loss of the proposed method and baseline. The baseline model is prone to over-fit all triplets in a one-to-one matching style. In contrast, the proposed method would converge to one non-zero constant. (b) Compatibility to existing methods, *i.e.*, Clip4Cir (Baldrati et al., 2022), on FashionIQ. *: We re-implemented the method with one ResNet50, and add our method. We also adopt a state-of-the-art cross-modality backbone Blip (Li et al., 2022), and add our method on the Shoes dataset. We observe that the proposed method could further improve the performance.

(a)

(b)

| FashionIQ | R@10 | R@50 |
|---|---|---|
| Clip4Cir* | 32.36 | 56.74 |
| Clip4Cir* + Ours | 34.19 | 59.23 |
| Shoes | R@10 | R@50 |
| Blip (Li et al., 2022) | 43.93 | 70.21 |
| Blip (Li et al., 2022) + Ours | 48.40 | 75.29 |

In contrast, our method (orange line) also converges but does not force the loss to be zero. Because we provide the second term in uncertainty regularization as Eq. 3, which could serve as a loose term.

**Only Coarse Retrieval Evaluation.** We design an interesting experiment only to consider query pairs with ambiguous coarse descriptions, *i.e.*, less than 5 words in the FashionIQ dress dataset (about 6,246 of 10,942 queries). Compared with the baseline model, our method improves $1.46\%$ Recall@10 rate and $3.34\%$ Recall@50 rate. The result verifies that our model can significantly improve coarse retrieval performance over baseline.

**Augment the source feature $f_s^I$.** We modify and add feature augments to the source image, but the Recall@10 rate decreases 1.43% and the Recall@50 rate decreases 2.09% on Shoes (see Table 3a). It is due to the conflict with the one-to-many matching. If we conduct the source feature augmentation, it will become to many-to-one matching. As the visual intuition in Figure 1b, it is better to apply such augmentation on the target feature instead.

**Complementary to other works?** Yes. We re-implement the competitive method Clip4Cir (Baldrati et al., 2022) in Table 4b, and show our method is complementary, further improving the recall. Similarly, we also adopt a state-of-the-art cross-modality backbone Blip (Li et al., 2022), and add our method to the Shoes dataset. We observe that the proposed method could further improve the performance of about $5\%$ on both Recall1@10 and Recall1@50.

## 5 CONCLUSION

In this work, we provide an early attempt at a unified learning approach to simultaneously modeling coarse- and fine-grained retrieval, which could provide a better user experience in real-world retrieval scenarios. Different from existing one-to-one matching approaches, the proposed uncertainty modeling explicitly considers the uncertainty fluctuation in the feature space. The feature fluctuation simulates the one-to-many matching for the coarse-grained retrieval. The multi-grained uncertainty regularization adaptively modifies the punishment according to the fluctuation degree during the entire training process, and thus can be combined with the conventional fine-grained loss to improve the performance further. Extensive experiments verify the effectiveness of the proposed method on the composed image retrieval with text feedback, especially in terms of recall rate. In the future, we will continue to explore the feasibility of the proposed method on common-object retrieval datasets, and involving the knowledge graph (Liu et al., 2020; Sun et al., 2021).

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
