# OpenReview forum: "Composed Image Retrieval with Text Feedback via Multi-grained Uncertainty Regularization"
_ICLR.cc/2024/Conference — ICLR 2024 poster_

### Official Review · Reviewer_cCoz · 2023-10-30

**Soundness:** 4 excellent
**Presentation:** 3 good
**Contribution:** 3 good
**Rating:** 8
**Confidence:** 5

**Summary:**

In this paper, the authors studied the image text composed retrieval in the field of fashion, aiming to prevent the model from prematurely excluding the correct candidate results in the early retrieval stage, and improve the recall of the retrieval task. To solve this problem, this paper proposed a composed retrieval model based on multi-granularity uncertainty regularization, which models coarse-grained and fine-grained retrieval simultaneously by considering multi-granularity uncertainty. The experiments showed that compared with existing methods, the proposed model can significantly improve the retrieval accuracy. This paper has strong practical significance and clear research motivation. However, there are also some problems in writing, which need to be strengthened.

**Strengths:**

a.	This paper found that in the process of image text combination query in the real world, the multi-round interaction process inevitably includes coarse-grained retrieval and fine-grained retrieval, and the traditional fine-grained metric learning method cannot meet the requirements of coarse-grained reasoning.
b.	In this paper, a new multi-granularity uncertainty regularization method is proposed. By uncertainty modeling and uncertainty regularization, fine-grained and coarse-grained matching can be learned during training. By controlling noise levels, the proposed method can be reduced to one-to-one metric learning, which is an extension of the traditional method.
c.	In this paper, a large number of experiments are carried out on three public data sets, and the experiments showed that the proposed method can improve the recall rate of existing methods.
d.	Since the proposed method is orthogonal to traditional methods, it can be combined with existing works to further improve their performance.

**Weaknesses:**

a.	Figure 1.a and section 3.1 did not clearly explain why there is uncertainty in triplet matching, and it is recommended to give examples of situations where the description is unclear or the source image is inaccurate.
b.	Since this paper mainly built the network using existing works [1][2], but did not describe their implementation, it is recommended to provide a more detailed explanation of the details of the network.
c.	There are some colloquial problems in the language of this paper, and some expressions need to be improved. It is suggested to polish the language.

Reference
[1]	Lee S, Kim D, Han B. Cosmo: Content-style modulation for image retrieval with text feedback[C]//Proceedings of the IEEE/CVF Conference on Computer Vision and Pattern Recognition. 2021: 802-812.
[2]	Chen Y, Gong S, Bazzani L. Image search with text feedback by visiolinguistic attention learning[C]//Proceedings of the IEEE/CVF Conference on Computer Vision and Pattern Recognition. 2020: 3001-3011.


Typos and minors
There are non-standard punctuation marks, such as P8, the third line of "Parameter sensitivity of the balance weight", etc. It is recommended to check the whole paper.

**Questions:**

please see the weaknesses.

---

> ### Author Response · Authors · 2023-11-22
> **Dear Reviewer cCoz**
>
> Dear Reviewer,
>
> Thank you for your thoughtful feedback on our paper regarding composed image retrieval with text feedback. We appreciate your detailed analysis and constructive suggestions.
>
> **Q1. Why uncertainty in triplet matching?**
>
> A: Thank you. We will modify the figures and provide more samples.
> Usually, we do not know whether the triplet annotation is exactly one-to-one, and there may exist many false-positive candidates.
> For instance, for coarse-grained matching, the short caption "change red and short sleeves" can match many different T-shirt candidates in the gallery;
> In contrast, we also have many fine-grained annotations in the dataset, which could exactly match one target object, such as, "change no sleeves with a black-white Micky mouse on the clothes".
> Therefore, facing such triplet annotation uncertainty, we intend to propose a method that could automatically learn multi-grained matching with flexibility.
>
> **Q2. Details of the network**
>
> A: Thank you. We will provide a more comprehensive description of the network architecture details in Figure 2, and provide more details in implementation details. In particular, we adopt the same compositor structure in the existing work CosMo for a fair comparison, which contains two modules: the Content Modulator (CM) and the Style Modulator (SM). The CM uses a Disentangled Multi-modal Non-local block (DMNL) to perform local updates to the reference image feature, while the SM employs a multimodal attention block to capture the style information conveyed by the modifier text. The outputs of the CM and SM are then fused using a simple concatenation operation, and the resulting feature is fed into a fully connected layer to produce the final image-text representation.
>
> We will add the above-mentioned details to the final version.
>
>
> **Q3. Language and punctuation**
>
> A: Thank you. We will refine expressions for clarity and formality, and conduct a comprehensive review to ensure proper punctuation marks throughout the entire paper.
>
>
> We really appreciate your supports and understanding.

---

### Official Review · Reviewer_FQ9t · 2023-10-31

**Soundness:** 2 fair
**Presentation:** 2 fair
**Contribution:** 2 fair
**Rating:** 3
**Confidence:** 4

**Summary:**

This paper identifies a training/test misalignment between the fine-grained metric learning and the demand on coarse-grained inference in the real-world image retrieval with text feedback. This paper introduces a new unified method to learn both fine- and coarse-grained matching during training. In particular, we leverage the uncertainty-based matching, which consists of uncertainty modeling and uncertainty regularization.

**Strengths:**

1. The application is interesting, matching source image and text with target image.
2. The paper is overall well-written and easy to follow.2
3. The designed method is reasonable.
4. The experimental results look good.

**Weaknesses:**

1. The technical novelty and contribution are limited. It is trivial that adding noise to features can enhance the robustness. The other components in this method are trivial.
2. The qualitative results are insufficient. The authors should provide more qualitative results, and provide in-depth analyses on the cases where the proposed method outperforms or underperforms the baselines.

**Questions:**

1. Highlight the technical contributions.
2. Provide more qualitative results and analyses.

---

> ### Author Response · Authors · 2023-11-22
> **Dear Reviewer FQ9t**
>
> Dear Reviewer,
>
> Thank you for your thoughtful feedback on our paper regarding composed image retrieval with text feedback. We appreciate your detailed analysis and constructive suggestions.
>
> **Q1. Technical novelty and adding noise is too simple.**
>
> A: Thank you.
>
> (1) Simple is our advantage. It is worth noting that the performance improvement we achieved, especially in Recall@50, is nontrivial. On the three public datasets, i.e., FashionIQ, Fashion200k, and Shoes, the proposed method has achieved +4.03\%, + 3.38\%, and + 2.40\% Recall@50 accuracy over a strong baseline, respectively. In this paper, we propose a new unified method to learn both fine- and coarse-grained matching during training. Additionally, our method is complementary, and when combined with SOTA, it can further enhance overall performance.
>
> (2) Adding noise is not an arbitrary idea. Actually, the noise assumption is designed in Eq.1 according to uncertainty. We derive the final uniform loss in Eq.5, and show the conventional fine-grained loss is a special case of Eq.5.
>
> **Q2. Insufficient qualitative result?**
>
> A: We appreciate you feedback regarding the qualitative results and the need for a more comprehensive analysis of the cases where our proposed method outperforms or underperforms the baselines. We have provided qualitative results in Figure 3. Can the reviewer specify what additional results of our experiment are needed? We are willing to provide them in the final version.

---

### Official Review · Reviewer_oWB8 · 2023-11-01

**Soundness:** 4 excellent
**Presentation:** 3 good
**Contribution:** 3 good
**Rating:** 8
**Confidence:** 4

**Summary:**

This paper proposes a novel approach to simultaneously model the coarse- and fine-grained retrieval by using multi-grained uncertainty, which contains uncertainty modeling and uncertainty regularization. The corresponding experiments validate the effectiveness of the proposed method on three public datasets.

**Strengths:**

1.The authors propose a new learning strategy in this paper, which is well-written and easy to follow.
2.This paper provides comprehensive experimental analysis and ablation discussion.

**Weaknesses:**

1.	The novelty is not well explained, please explain the difference from existing methods. The authors have to provide convincing proof to show why their proposed learning strategy is significantly different from the former methods. In Table 4(b), the author has provided a very simple discussion and experiment. However, it would be beneficial to include additional comparison methods and provide a more detailed discussion.
2.	In the method section of this paper, it is important to highlight the unique aspects of your proposed learning strategy. While the uncertainty modeling approach presented is easy to follow, it should be acknowledged that its simplicity may also be considered a limitation in terms of novelty.
3.	The comparison baseline of this paper is not sufficiently novel, consider exploring alternative methods that may be more unique. Recent three years of state-of-the-art methods should be included in the comparison experiment section to validate the effectiveness of the proposed method.
4.	As mentioned, the main contribution of this paper includes three points. The description of the first contribution is not clear enough.
5.	The tables in this paper are informative, but several of them require further standardization and uniformity.

**Questions:**

None

---

> ### Author Response · Authors · 2023-11-22
> **Dear Reviewer oWB8**
>
> Dear Reviewer,
>
> Thank you for your thoughtful feedback on our paper regarding composed image retrieval with text feedback. We appreciate your detailed analysis and constructive suggestions.
>
> **Q1-1. Explain novelty. The difference from existing methods with convincing proof**
>
> A: Thank you. Existing methods merely focus on the fine-grained search, by harnessing positive and negative pairs during training. This pair-based paradigm only considers the one-to-one distance between a pair of specific points, which is not aligned with the one-to-many coarse-grained retrieval process and compromises the recall rate.
> In an attempt to fill this gap, we introduce an uncertainty learning approach to simultaneously modeling the coarse- and fine-grained retrieval by considering the multi-grained uncertainty.
> It is worth noting that the proposed method is focused on better learning objectives, which is complementary to existing works on compositor design.
>
> **Q1-2 / Q3. The baseline selection? More additional comparison methods and a more detailed discussion in Table 4(b). More SOTA methods.**
>
> A: Thank you. Considering the availability of open-source code, we select a widely-used method, CosMo, as baseline. We also show the scalability of our method to a recent baseline, Clip4Cir in Table 4(b).
> In Table 1, we have compared ARTEMIS and DCNet, both within the last three years.
> We will further add FashionViL and Pic2Word in comparison as follows.
>
> | Average on FashionIQ | R@10      | R@50      |
> | -------------- | ----- | ----- |
> | ARTEMIS [1] | 26.05  | 50.29 |
> | DCNet [2] | 27.78 | 53.89  |
> | Pic2Word [4] | 24.70   | 43.70   |
> | FashionViL [3]    | 26.60 | 52.74 |
> | FashionViL* [3]  | 31.21 | 57.04  |
> | Ours   | **33.17**  | **61.39** |
> (* Note that FashionViL introduces extra datasets for training. )
>
> [1] Delmas et al. Artemis: Attention-based retrieval with text-explicit matching and implicit similarity. In ICLR, 2022.
>
> [2] Kim et al. Dual Compositional Learning in Interactive Image Retrieval. In AAAI, 2021.
>
> [3] Han et al. FashionViL: Fashion-Focused Vision-and-Language Representation Learning. In ECCV, 2022
>
> [4] Saito et al. Pic2Word: Mapping Pictures to Words for Zero-shot Composed Image Retrieval. In CVPR, 2023
>
> Besides, as suggested, we adopt a strong cross-modality baseline BLIP [5], and add our method. We observe that the proposed method could improve the performance as follows.
>
> | Shoes | R@10      | R@50      |
> | -------------------- | --------- | --------- |
> | BLIP  | 43.93 | 70.21 |
> | BLIP + Ours   | 48.40 | 75.29 |
>
> [5] Li et al. BLIP: Bootstrapping Language-Image Pre-training. In ICML 2022
>
> **Q2. Highlight the unique aspects in Method Section. Too simple?**
>
> A: Thank you. One unique aspect is Eq. 5, which uniforms the fine-grained and coarse-grained learning. In particular, the fine-grained loss can be viewed as a special case of coarse-grained loss, whose variance is a constant term. In this way, we could unify the multi-grained learning losses and easily balance them via a dynamic weight $\gamma$. We also have a detailed discussion at the end of Method Section on page 6.
>
> Yes.  Simple is also our advantage. Despite its straightforward nature, our approach brings about significant performance improvements. On the three public datasets, i.e., FashionIQ, Fashion200k, and Shoes, the proposed method has achieved +4.03%, + 3.38%, and + 2.40% Recall@50 accuracy over a strong baseline, respectively.
> This simplicity also ensures broader accessibility and applicability in various contexts. We hope that the simple method can serve as a new baseline and give the community a new perspective.
>
> **Q4. The description of the first contribution.**
>
> A: Thank you. In the first contribution, we hope to point out a long-exist misalignment problem, which compromises the training. The problem is due to the multi-grained annotation of the datasets. For instance, "change red" is a coarse-grained feedback, while "change red and short sleeves with a blue white Micky mouse on the clothes" is a fine-grained feedback. It is challenging to learn the multi-grained by the conventional one-to-one metric learning losses, like infoNCE loss, sphere loss and etc. We will modify the first point of motivation as follows:
>
> *We pinpoint a training/test misalignment in real-world image retrieval with text feedback, specifically between fine-grained metric learning and the practical need for coarse-grained inference. Traditional metric learning mostly focuses on one-to-one alignment, adversely impacting one-to-many coarse-grained learning. This problem identification underscores the gap we address in our approach.*
>
> **Q5. The standardization and uniformity of table**
>
> A: Thanks a lot for your suggestion. We will re-arrange the table location and modify the table style to make its style more uniform.

---

### Author Response · Authors · 2023-11-22
**Overall**

We thank the reviewers for their constructive comments and suggestions. We are encouraged that they found our motivation and idea are **well-written and easy to follow** (RoWB8, RFQ9t), **reasonable** (RFQ9t),  **orthogonality to traditional approaches and potential for combination with existing methods** (RcCoz). Our experimental analysis and ablation discussion are **comprehensive** (RoWB8). **The experimental results look good** (RFQ9t, RoWB8).
Due to the page limitation, we will fix all other minor points in the revised manuscript. Thanks a lot for your understanding.

---

### Meta-Review · Area_Chair_VxQu · 2023-12-14

**Metareview:**

This paper focuses on composed image retrieval with text feedback, and present to simultaneously model both coarse-grained and fine-grained retrieval according to multi-grained uncertainty. Specifically, the approach includes two uncertainty strategies from modelling and regularization from two perspectives: 1) introducing identically distributed fluctuations in the feature space for modelling uncertainty; 2) incorporating uncertainty regularization to adapt the matching objective according to the fluctuation range. Extensive experiments on three public datasets demonstrate the effectiveness compared to previous methods.

Strengths: This paper found traditional fine-grained metric learning fails to enable coarse-grained reasoning, and then presents a multi-granularity uncertainty modeling and regularization to simultaneously learn fine-grained and coarse-grained marching. In this paper, extensive experiments on the public sets show the effectiveness, and the proposed method is orthogonal to traditional approaches.

Weaknesses: The novelty is not well explained and the baseline comparison is not novel enough.

**Justification For Why Not Higher Score:**

The baseline methods are majorly old except for FashionViL that is added in the rebuttal phrase.

**Justification For Why Not Lower Score:**

The paper presents a good finding that previous approaches fail to achieve coarse-grained reasoning, and introduces effective strategies to facilitate composed image retrieval via multi-grained uncertainty modelling and regularization.

---

### Decision · Program_Chairs · 2024-01-16

Accept (poster)